# Improving on-field decision making using video-based training – A pilot study with young volleyball players

Silke De Waelle[1⊙], Simon J. Bennett [ID][2‡*], Mark A. Scott[2‡], Matthieu Lenoir[1‡], Frederik J. A. Deconinck [ID][1⊙]

1 Department of Movement and Sports Sciences, Ghent University, Ghent, Belgium, 2 School of Sport and Exercise Sciences, Liverpool John Moores University, Liverpool, United Kingdom

⊙ These authors contributed equally to this work.
‡ SJB, MAS and ML also contributed equally to this work.
* s.j.bennett@ljmu.ac.uk

## Abstract

Decision making is a crucial skill in several sports, which researchers and practitioners have for a long time sought to improve. Recent studies have focused on the potential of video-based training, but most examined adult athletes and did not typically measure transfer to on-field performance. Therefore, we investigated the effect of a 4-week video-based training on youth volleyball players' on-field decision making. Twenty female volleyball players between 13 and 15 years old were initially assigned to either an experimental or control group, of whom 16 (eight per group) completed the entire study. Decision making skill was assessed on a video-based task, as well as an on-field task. The results of our pilot study demonstrated that only the experimental group significantly improved their decision making skills on both the video-based assessment and on-field test (motor decision) between baseline and retention. This pilot study indicates that video-based training can be an effective way to improve young players' on-field decision making skills, although improvements in motor execution may require intermittent physical practice.

## Introduction

To make appropriate decisions in team sports such as soccer, volleyball or basketball, it is necessary to process and monitor multiple sources of information from opponents, teammates and the ball [1]. Expert athletes in these sports exhibit superior decision making skills than sub-elite athletes or novices [2,3]. This expert advantage is even present in young athletes (i.e., during adolescence), with high-level soccer players outperforming lower-level athletes on decision making skills [4]. The implication is that decision making is already an important aspect of expertise in

**Data availability statement:** All relevant data are within the manuscript and its Supporting information files.

**Funding:** The author(s) received no specific funding for this work.

**Competing interests:** The authors have declared that no competing interests exist.

young athletes, and that its development at this crucial period for talent identification warrants consideration [5,6].

On-field decision making in team sports can be trained by manipulating task constraints that influence the number or nature of decisions to be made [7,8]. This is often done by means of small-sided games, which usually involve fewer opponents and a smaller pitch size in order to reduce the number of decision making options. Despite the success of this on-field training method, it is not simple and/or efficient to implement, and is not individualized. Therefore, researchers have considered how to improve decision making through off-field training [9], such as watching sport-specific videos outside of their regular practice sessions. Typically, this would involve watching an unfolding sequence of play, with the video being stopped at a key point of interest (e.g., ball-foot contact in soccer), after which the participant is required to select the best possible decision (e.g., where to pass the ball) by means of a button press, verbal report or the appropriate sport-specific movement [10].

In sports such as softball [11], cricket [12] and basketball [13], most studies have shown task-specific improvements in adult participants decision making after sports-specific video-based training. However, the extent to which the positive effects of off-field, video-based training transfer to on-field performance remains unclear [14], in part because transfer is rarely measured [15,16]. Of the few studies that did examine transfer, the evidence is equivocal [12,13,17,18]. For example, Gorman and Farrow [18] found that a four-week, video-based training intervention in basketball improved decision making of all groups of adult participants (including the control and placebo group) on the video-based test, but not in on-field, match situations. Conversely, Pagé and colleagues [13] reported transfer of video-based training effects to on-field decision making in basketball, but only when the on-field situations matched those presented during the training (near transfer). Moreover, when practice occurred in virtual reality using a head mounted display instead of 2D videos displayed on a computer monitor, the training effect was extended to untrained situations (far transfer).

With youth athletes there are a similarly small number of studies that have investigated the effect of video-based training on decision making skills [17]. As with the adult studies, findings are mixed and little consideration is given to on-field transfer. Nimmerichter and colleagues [14] reported that 6 weeks of video-based training improved the video-based decision making skills of 14-year-old soccer players, but transfer to on-field performance was not measured. Panchuk and colleagues [18] found that 17-year-old female basketball players (n=6) significantly improved their decision making following 3-weeks of immersive training on a video-based decision making test, but only to a similar extent as the control group (n=3) (15). The male counterparts showed no improvements in video-based decision making irrespective of whether they took part in the intervention (n=5) or control protocol (n=4). No significant improvements were found in on-field performance for either group. More positive effects initially seem apparent in a recent meta-analysis on the effectiveness of decision making training in volleyball players [19]. However, closer inspection of the six original articles that studied youth players indicates that only four used off-field video-based training, and that of those only one found a 20% positive transfer to on-field performance [20].

At present, therefore, despite there being some evidence that video-based training can be a useful tool for the improvement of video-based decision making abilities, the extent of transfer to on-field performance remains unclear, particularly in youth athletes. The current pilot study aims to explore the effects of a four-week, video-based decision making intervention on on-field performance in youth female volleyball players. As commented above, decision making in volleyball requires the athlete to process and monitor multiple sources of information from opponents, teammates and the ball. The constant movement of the ball (i.e., it is not allowed to be caught and held) and the speed at which it travels (e.g., in excess of 100 km/h), combined with the dynamic player positioning, means that quick, accurate decisions on where and how to spike the ball are essential to capitalize on scoring opportunities. Simple and efficient training methods that improve decision making of youth athletes could have meaningful impact on their enjoyment and progression, as well as match outcome. To assess learning and transfer in these athletes, baseline and retention tests on video-based and on-field decision making tasks in volleyball were performed either side of the training phase. To permit a comprehensive analysis of on-field decision making [21], three measures were included: verbal decision, motor decision and motor execution. The general hypothesis was that the experimental group would improve their decision making performance on both the video-based task and on-field decision making task, while we expected no improvements in the control group on either task.

## Materials and methods

### Participants

Assuming an effect size of Cohen's f = 0.25 ($\eta p^2 = 0.059$), a power of 0.8 and a correlation between repeated measurements of 0.25, it was determined that a total sample of N = 50 would be required for the interaction effect of our mixed design ANOVA (G*Power 3.1). Despite only being able to recruit 20 participants, it was decided to proceed with this sample on the basis that the initial effect size estimation could have been too conservative compared to previous similar work [12,13,18], and that any findings could still contribute to subsequent meta-analysis.

The recruitment period started on 03/08/2020 and ended on 25/09/2020. Participants of the under 15 age category were recruited from four different teams within three Flemish youth volleyball clubs and were either assigned to the control group that would be receiving the placebo training (n = 9), or the experimental group that would be receiving the decision making training (n = 11). To limit baseline difference between groups, the selected volleyball teams were all playing at the same competition level. Participants were unaware of the fact that there were two different groups, and were randomly assigned to the experimental or control group per team. Prior to the study, participants and their parents provided written informed consent and were made aware of the fact that they could withdraw from the study at any time without consequences. The study was conducted in accordance with the ethical standards of the Helsinki Declaration and approved by the Ghent University Hospital ethics committee (registration number: B670201836811).

### Procedure

Participants completed baseline assessments of video-based and on-field decision making skill, followed by a 4-week intervention involving different types of video-based training. The initial study design included measurements of both the video-based and on-field tests immediately following the intervention. However, it was only possible to administer the video-based test at the participant's home without the supervision of a trained researcher (NB. These data are not reported). Between 28 and 31 days after the intervention ended, participants completed retention assessments of both the video-based and the on-field tests. Neither of the groups took part in any on-field practice during the study.

### Measurements

**Video-based test.** The video-based decision making task was based on the decision making task developed by De Waelle and colleagues [4] and validated for use in youth volleyball players. In this task, players viewed a series of 38

video clips of a progressing offensive sequence that was occluded 66ms before ball contact by the spiker. Participants were asked to imagine themselves as the spiker, and to decide on the most optimal zone to play the ball (i.e., with the highest chance of scoring a point). They then indicated their decision by pressing the corresponding key on a keyboard as fast as possible. To ensure face validity of the task, the video scenarios were developed in close collaboration with three certified volleyball coaches. The test included four different viewing conditions with 2, 3, 4 and 6 opponents, thereby reflecting realistic volleyball training and match situations. For the first two conditions, with 2 and 3 opponents, participants could choose between 6 zones, while for the last two conditions (with 4 and 6 opponents), there were 9 zones to choose from. The inclusion of more players resulted in progressively more complex conditions requiring a larger field size and more zones. This also ensured that zone size was similar across all conditions. The number of opponents, the size of the field, the number of zones and the number of clips per condition are displayed in Table 1. Each video clip lasted about 5 seconds, after which participants had 5 seconds to execute their response. However, participants did not have to wait until the screen was occluded to make their decision. A screenshot of exemplar video clips is presented in Fig 1, and a schematic of the test setting is shown in Fig 2.

The test took place in a separate and quiet room, with only the participant and the experimenter present. The video clips were projected on a 1.07m (w) x 0.6m (l) projection screen using an LED HD video projector (LG PH550G, Seoul, South Korea) that was placed on a table 1.50m from the screen. Subjects were instructed to stand behind this table at 2.00m from the screen. Video clips were displayed with OpenSesame software [22], which was also used to record the responses using a regular USB-connected keyboard.

**Table 1. Characteristics of the different viewing conditions in the decision making task.**

| Viewing Condition | Number of opponents | Field size (LxW in m) | Number of zones | Number of clips |
|---|---|---|---|---|
| 1 (2x2) | 2 | 6 x 4.5 | 6 | 10 |
| 2 (3x3) | 3 | 6 x 6 | 6 | 10 |
| 3 (4x4) | 4 | 7 x 7 | 9 | 8 |
| 4 (6x6) | 6 | 9 x 9 | 9 | 10 |
| Total | | | | 38 |

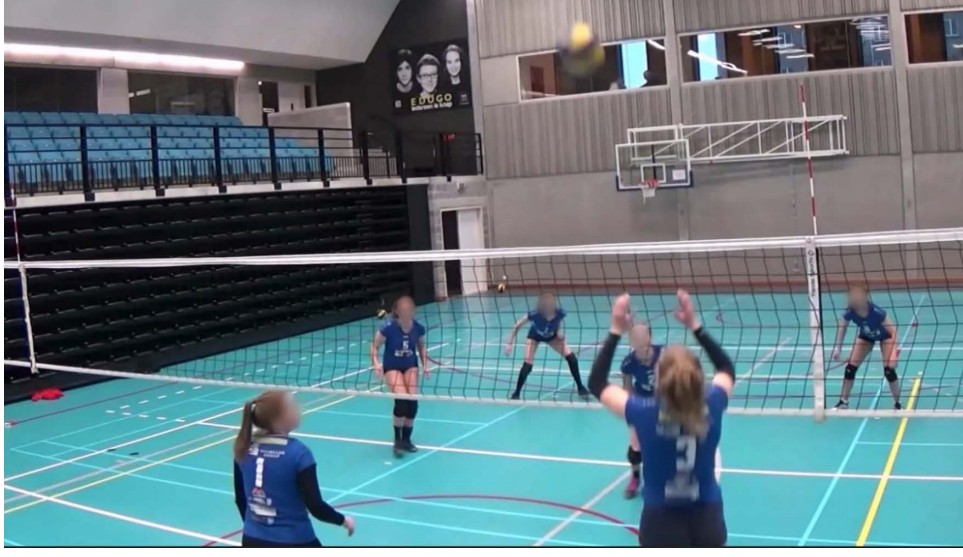

**Fig 1. Screenshot of the video-based decision making test.**

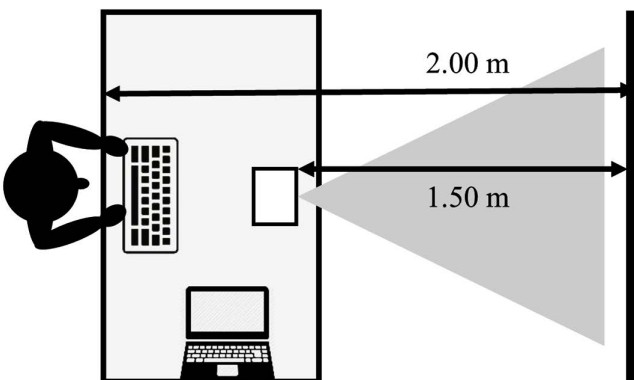

**Fig 2. Schematic of the video-based test setting.**

At the start of the decision making test, participants received detailed instruction, as well as an example and two practice trials. All participants completed the different viewing conditions in the same order, and within each condition trials were randomized for each participant separately. For each new viewing condition, a familiarization clip was shown, as the number of players and size of the court would change. The decision making test lasted about 10 minutes.

**On-field test.** A series of simulated offensive sequences using actual players that represented the video-based test was used for the on-field decision making test, but only with six opponents as this relates closely to the actual game setting. Eight experienced volleyball players were recruited as 'actors' to play the roles of the six defenders on the defensive side, as well as the libero (who plays the ball towards the setter) and setter (who passes the ball to the participant) on the side of the participant. This did not reproduce exactly the video-based test but it did ensure that the participants received high-quality passes that provided them with optimal scoring conditions in terms of time and space. Before the test, the six defenders were given explanations and time to practice the different defence patterns they would have to execute. These defence patterns were the same as in the video-based test, and prior to each trial, one of the researchers would indicate through a sign which defence pattern was to be executed. The defence patterns were presented in a random order that was the same for all participants. The field of the defenders was divided in 9 zones using orange tape and zones were marked with colours to facilitate the participant's verbal response. The net was a regulation height of 2.24m and the ball was a Mikasa V020W size 5. A schematic is provided in Fig 3.

Prior to the test the participant was given time 15 minutes to warm-up, including light jogging, dynamic stretches and volleyball-specific passing and attacking drills. Each trial of the test started with a free ball tossed to the libero on the participant's side. The libero would then play the ball to the setter, who would pass the ball to the participant acting as the spiker. The participant was instructed to play the ball where they were most likely to score on the opposing side of the field, using any technique. They were also asked to call out the zone where they intended to play the ball (i.e., verbal reports). This was done because participants between 13 and 15 years old are not always technically skilled enough to play the ball where they actually intend to play, and thus verbal reports provide information on intended behaviour. If the defenders were able to defend the ball, they played a free-ball back to the libero on the participant's side to start a new trial. If the ball was not defended, a new trial was started by a free-ball toss to the libero. Verbal reports were noted on paper by one of the researchers at the time of testing, and two cameras (see Fig 3) recorded the test from different angles to allow for analyses of (1) each decision, based on the landing location of the ball as well as the execution of the action, and (2) the correctness of the actors' defence positions. Participants completed 20 trials as the left wing spiker and 20 trials as the right wing spiker. Rest periods between trials were not included, other than the time it took to prepare for the next trial, which reflects an actual game situation. The entire on-field test lasted 15–20 minutes, and there were no

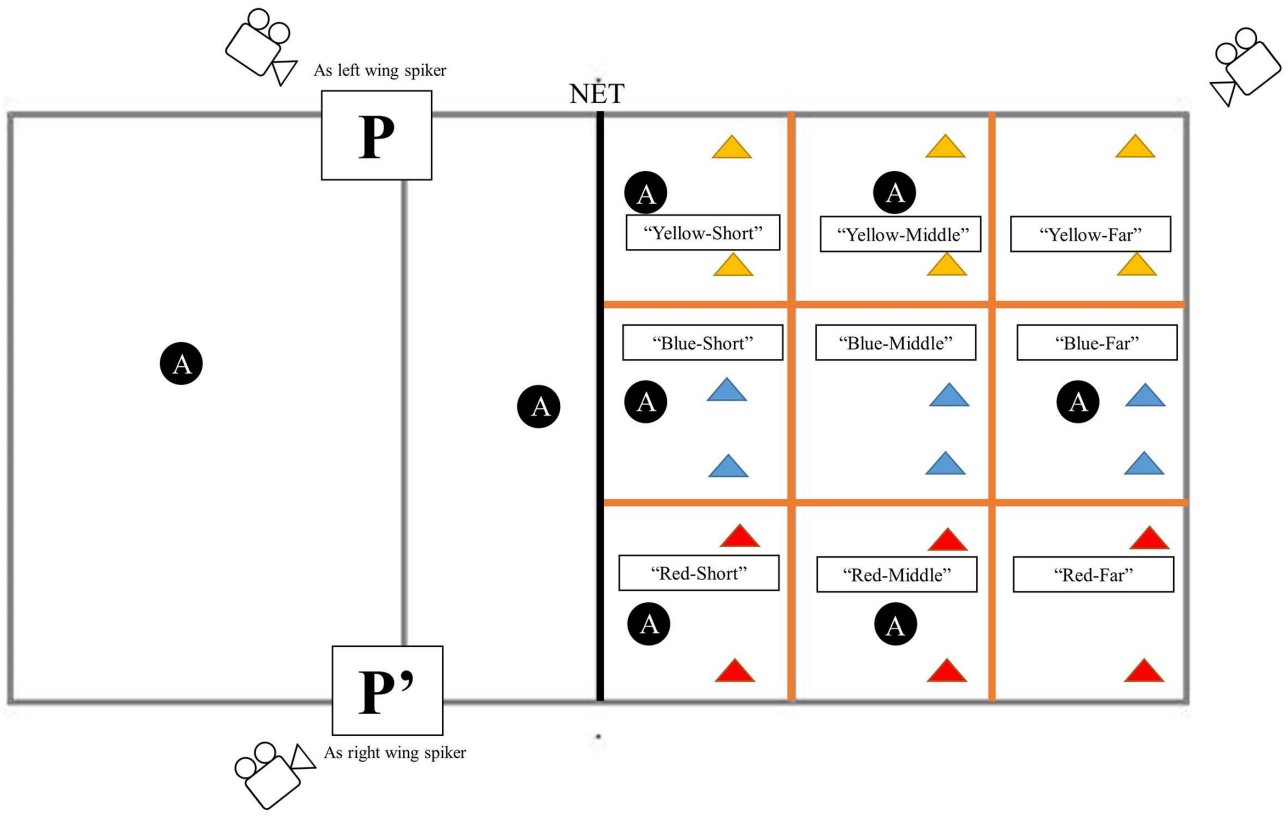

**Fig 3. Schematic of the set-up of the on-field test.** P = participant position, P' = alternative position of the participant, A = Actor.

obvious signs or reports of fatigue from the participants. While a participant was actively performing the on-field task, no other participant were present.

### Intervention

Both groups received 4 weeks of video-based training, with 4 sessions of 15 minutes per week, resulting in a total of 16 training sessions. The training sessions were conducted at home using the OpenSesame software, and commenced during the first week of the 2020–2021 season. Participants received a detailed manual before the study and were contacted by one of the research team to check that the software had been correctly installed and that they understood when and how to use it. Participants were instructed to spread out their training sessions over the week, and to not perform two sessions during the same day. This was checked in the log-files created by the OpenSesame software, which the participant sent by email to the research team after each training session. The log-files contained information about the participants' responses and response times during training, as well as when they executed each session. While it is feasible that someone other than the participant could have completed the training, it is highly unlikely given that they would need access to the participants computer and email.

**Experimental group.** The experimental group received training sessions that were highly similar to the video-based test, with the same instructions and manner of response. However, training sessions became gradually more difficult, starting with 2 opponents in the first week, 3 and 4 opponents in the second week, and ending with 6 opponents during the last two weeks. At the beginning of each training session, detailed instruction was given about which areas of the

display players should pay attention to in order to make optimal decisions. Different topics were addressed within each of the difficulty levels. For example, it was explained that the role of the setter is to pass the ball to the left wing or the right wing spiker to ensure they have the time and space to score a point, and that the defending team will try to block the ball played by the spiker, usually with two players at the net in a position relative to each other that limits the zones available to the spiker that are not covered by remaining four defenders. It was also clarified what to do when receiving a pass that is difficult for the spiker to reach, or does not provide optimal scoring conditions. In this way, the intervention aimed to improve the participants' decision making in standard, but also in non-standard situations (e.g., when you get a bad pass, or when the block has not formed correctly). Per training session, each participant completed 20 trials, i.e., 20 different video clips of 20 different volleyball scenarios. They were provided with feedback after each trial about the outcome of their response.

**Control group.** The structure of the placebo training was similar to the experimental group, as they too completed 16 sessions (4 weeks x 4 sessions/week) that included detailed instructions followed by video clips to which they had to respond using the keyboard. The clips and responses were designed such that participants engaged in game settings. In the first two weeks, participants had to judge the quality of the first ball reception as quickly as possible, and in the third week, they had to judge the quality of the pass given by the setter. In the last week, they had to predict the direction of set-up. Accordingly, these tasks required participants to focus on the ball's flight path, and not the positioning of blockers and defenders, which is considered critical for offensive decision making. As the control group did not receive feedback between trials, they completed 30 trials each training session to ensure they spent a similar amount of time performing the training task.

## Dependent variables

**Video-based decision making.** Prior to the study, a panel of three expert coaches viewed the video clips for the decision making test and selected the most optimal zone (i.e., the zone with the highest scoring probability, based on the position of the block and the defenders). Only the clips for which all three experts agreed were included in the test. Each participant's performance on the video-based decision making task was assessed using the percentage accuracy of the 6x6 condition (i.e., percentage of times the optimal zone was selected). The 6x6 condition was chosen because this relates most closely to the on-field test and to the actual game that is played by this age group.

**On-field performance.** This was assessed using 3 separate measures. First, the *on-field verbal decision*, i.e., where participants said they were going to play, was registered. Second, the *on-field motor decision* was recorded, i.e., where participants intended to play the ball, regardless of the quality of their execution and final ball landing location [21]. To this end, two experienced volleyball coaches analysed the direction in which the ball was played, in combination with the technique used, to decide on the player's intended action. Interrater reliability of this judgement was examined by having 2 coaches evaluate all trials (n = 40) from 4 participants (2 from each group). This analysis indicated excellent agreement (Cohen's unweighted kappa = 0.814). Finally, the *on-field motor execution*, i.e. what participants ended up doing, for which actual ball landing location was analysed from the video footage from the on-field test. For each of the three measurements, percentage accuracy was calculated per participant for the baseline and retention test.

## Data analysis

Of the initial 20 participants, three from the experimental group dropped out during the study as they found it too time-consuming. In addition, one participant from the control group was unable to complete the retention test. Therefore, for both the video-based tests and on-field tests, we had complete data sets (baseline and retention) for 8 participants in each group (see Table 2).

To assess whether the intervention influenced learning of the video-based decision making task, the percentage accuracy scores were submitted to a mixed ANOVA, with group as the between-subjects factor and phase as the within-subjects factor (baseline, retention). To assess learning of the on-field task, the three dependent measures (verbal decision, motor decision and motor execution) were submitted to a mixed MANOVA, followed by mixed ANOVA, with group as the between-subjects

**Table 2. Summary of participant demographics.**

|  | Experimental | Control |
|---|---|---|
| **Age** | 13.0 (1.0) | 13.6 (0.6) |
| **Volleyball Experience** | 4.4 (2.4) | 5.3 (1.3) |
| **N** | 8 | 8 |

factor and phase (baseline, retention) as the within-subjects factor. Assumptions of normality and homogeneity of variances were confirmed using QQ plots and Levene's test. Significant interaction effects were further analysed using pairwise comparisons of estimated marginal means with Bonferroni correction. The alpha level was set to 0.05 and partial eta square effect sizes are reported. Statistical tests were conducted using IBM SPSS Statistics for Windows, Version 28.0.

## Results

### Video-based decision making task

Mixed ANOVA indicated a main effect for phase, $F_{(1,14)} = 10.93$, $p = 0.005$, $\eta_p^2 = 0.44$, as well as a significant phase x group interaction, $F_{(1, 14)} = 9.43$, $p = 0.008$, $\eta_p^2 = 0.40$. There was no significant main effect of group, $F_{(1, 14)} = 0.69$, $p = 0.42$, $\eta_p^2 = 0.05$. While performance of the control group did not differ between baseline and retention (Fig 4B), there was a significant improvement ($p = 0.003$) exhibited by the experimental group (Fig 4A). Importantly, no between-group differences were apparent at baseline.

### On-field task

Mixed MANOVA revealed no significant multivariate main effect for phase, $F_{(3, 12)} = 1.29$, $p = 0.332$, $\eta_p^2 = 0.22$, or group, $F_{(3, 12)} = 1.89$, $p = 0.185$, $\eta_p^2 = 0.32$, but there was a significant multivariate phase x group interaction, $F_{(3, 12)} = 4.65$, $p = 0.022$, $\eta_p^2 = 0.54$. Univariate tests also indicated no significant main effect for phase or group for any of the dependent variables. However, there was a significant phase x group interaction for verbal decision, $F_{(1, 14)} = 9.19$, $p = 0.009$, $\eta_p^2 = 0.40$, and

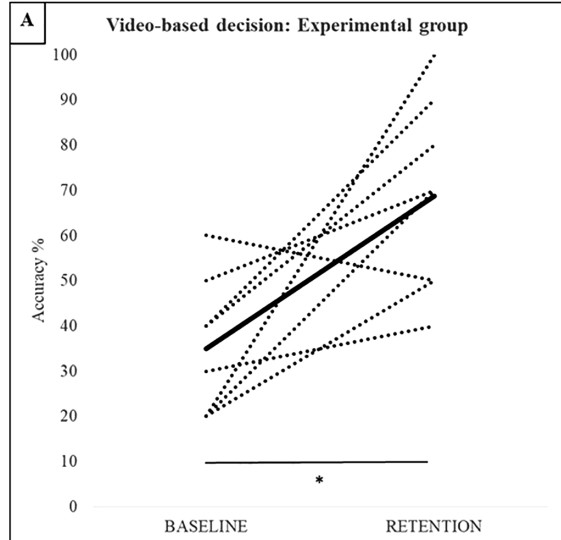
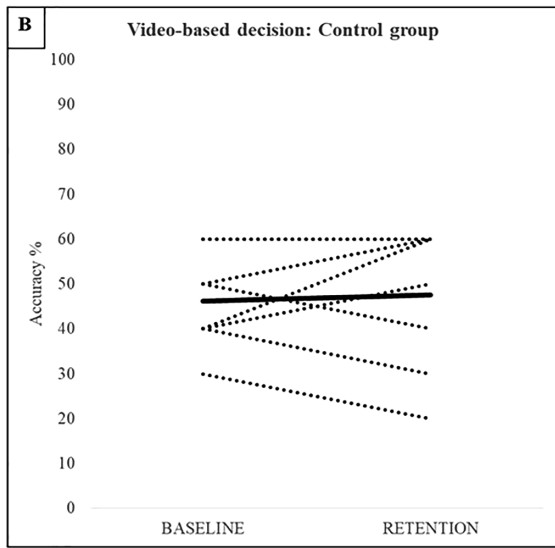

**Fig 4. Accuracy scores for the video-based decision making (6x6 condition).** Dotted lines represent individual results, the black line represents the group mean. *significant difference in the group mean, p < 0.05.

motor decision, $F_{(1, 14)} = 13.39$, $p = 0.003$, $\eta_p^2 = 0.49$, but not for motor execution, $F_{(1, 14)} = 0.631$, $p = 0.44$, $\eta_p^2 = 0.04$. There was no difference between the groups at baseline or retention for any of the dependent measures. However, the experimental group did significantly improve their motor decisions (Fig 5C, $p = 0.017$), but not their verbal (Fig 5A, $p = 0.085$) and

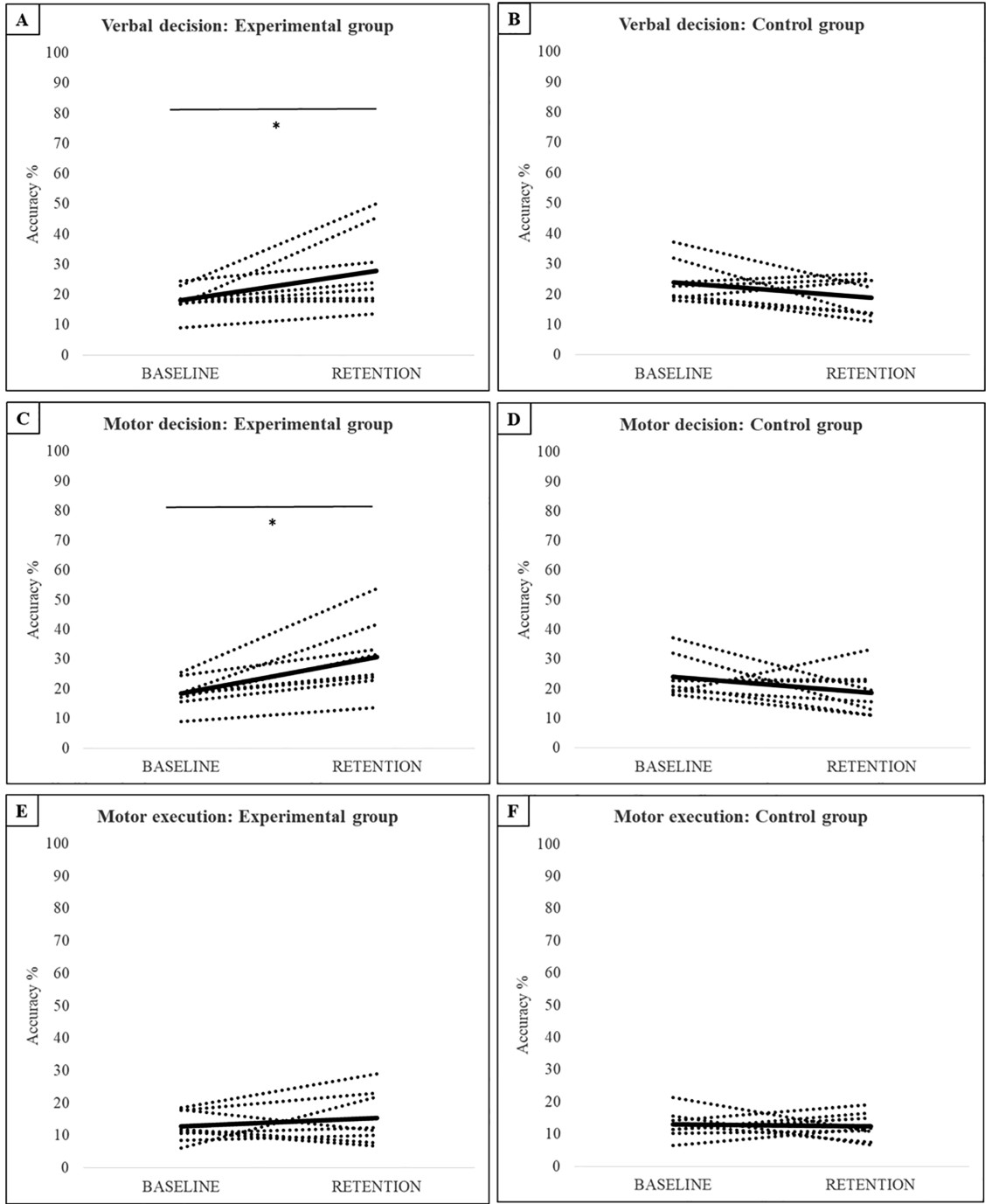

**Fig 5. Accuracy scores for the on-field decision making variables.** Dotted lines represent individual results, the black line represents the group mean. *significant difference in the group mean, p<0.05.

motor execution (Fig 5E *p = 1.00*), between baseline and retention. The control group did not show any improvement in their verbal decision (Fig 5B), motor decision (Fig 5D), or motor execution (Fig 5F), with negative group mean differences between baseline and retention for each measure.

## Discussion

This pilot study investigated the effect of a 4-week video-based training intervention on the decision making skills of youth volleyball players. Consistent with the general hypothesis, only the experimental group exhibited improved decision making in the video-based task, as well as the controlled and representative on-field task (motor decision only). This latter finding is in line with some previous studies [13], and suggests that video-based training can indeed be useful to improve decision making skills in youth team sports athletes. Contrasting effects have been reported [18,23] but there are notable differences in the methodologies. For example, the aforementioned studies measured on-field decision making in small-sided games or actual match performance (far transfer), whereas both the current study and Pagé et al. [13] used a controlled, on-field test designed specifically to replicate the situations of the video-based test (near transfer). In addition, we found evidence for improved on-field decision making skills when video-based training employed a first-person perspective and provided feedback after each trial (see also [13]), implying that these may be crucial elements for a successful video-based training intervention.

Although there was a significant interaction for verbal decision making in the on-field task, there was no improvement by the experimental group after Bonferroni correction for multiple pairwise comparisons. Verbalizing a motor plan is a common approach in perceptual-cognitive research, but is a somewhat artificial task that is rarely practiced in an on-field setting. This could have posed an additional challenge for the youth volleyball players in the current study, thereby limiting their improvement. Observation of the individual data (Fig 5A) shows that while none of the participants in the experimental group exhibited a decrease in verbal decision making accuracy from baseline to retention, 2 participants exhibited less than 1% change. The lack of improvement by the experimental group in on-field motor execution was more robust, with no significant main or interaction effects. Observation of the individual data (Fig 5E) shows that 3 of 8 participants in the experimental group exhibited a decrease between baseline and retention. This finding is partly consistent with a recent systematic review that showed programs designed to train decision making usually only benefit perceptual-cognitive measures, and not perceptual-motor measures [17]. This could simply be due to the fact that perception and action are uncoupled in video-based training, with no or a non-representative motor execution required when responding to the video clips. However, the lack of perception-action coupling in the current study did not prevent participants from improving on the motor decision task, indicating that they were better able to correctly identify where and how to respond following the intervention. An alternative interpretation could be that not having the opportunity to physically practice the motor task, such as during 2 months of the current intervention, prevented participants from applying what they learned during the video-based training. Similar effects are found in work on voluntary imitation, with observational practice alone being less beneficial to subsequent motor execution compared to observational practice interspersed with physical practice [24,25]. As is the case for observational practice [26], it is possible that video-based training enables participants to perceive and represent relevant information, but it does not engage feedforward and feedback processes that are critical for skilled motor execution. The implication for future studies on decision making in sport is that video-based training should be combined with physical training, although the frequency and volume of each type of training remains to be determined. That said, if an individual is unable to take part in on-field practice sessions, video-based training can provide a valuable alternative by which decision making skills, and in particular motor decision making, can still be improved. Indeed, when looking at the individual data (Fig 5C and 5D), it can be seen that the majority of participants in the control group showed a decline or no change in performance, whereas all participants in the experimental group showed an increase in performance.

This implies that video-based training might also be important in avoiding performance decline when athletes cannot physically train, for example when they are injured [27].

While this pilot study provides valuable insights into the use of individualized video-based training, it is important to acknowledge that there are some limitations. As noted in the methods, we initially planned to recruit 50 participants, but were only able to recruit 20 participants, of whom 4 did not complete the retention test, leaving us with 8 participants per group. Although this meant a reduction in power to detect smaller and moderate effects, a post-hoc sensitivity analysis indicated that this sample size was sufficient to detect an effect size of at least $\eta_p^2 = 0.175$ with a power of 80%. Notably, the effect size of $\eta_p^2 = 0.04$ found for motor execution was lower than both the a-priori and post-hoc critical thresholds, thus indicating that the finding of no significant improvement in this dependent measure by the experimental group was not due to insufficient power. As with some previous studies [20,28], the small sample size not only highlights the difficulty recruiting participants for training studies involving several sessions over several weeks (16 in total) followed by a delayed-retention, in which there are measures of off-field and on-field performance, but also raises a question about generalizability of the findings and the need for a replication study with a larger sample. Ideally, this would include a longer term follow-up to determine if there is any consolidation of the learning effects having resumed typical practice procedures. It could also be interesting to include more sequences of offensive play to better reflect the dynamic complexity of real competitions, and to determine transfer to the field in actual volleyball matches. Indeed, our use of simulated 6x6 offensive sequences with actual players in the on-field test was intended to closely reproduce the video-based test, and as such did not allow us to determine transfer to novel attacking sequences. Conducting such a study would benefit from collaboration with coaches of larger teams, allowing decision-making training to be integrated into regular practice activities, while also reducing the risk of participant drop-out.

In conclusion, this pilot study provides preliminary support for the effectiveness of video-based training for improving and retaining on-field motor decision making in youth volleyball players. Although the quality of motor execution did not improve, we suggest that this could be overcome by providing intermittent on-field training. Video-based training is easily accessible and low cost compared to virtual reality training, and could thus be an effective tool to maintain or improve the decision making abilities of youth players from grassroots to club level.

## Supporting information

**S1 File. Inclusivity in global research.**
(DOCX)

**S2 File. Raw data.**
(XLSX)

## Author contributions

**Conceptualization:** Silke De Waelle, Simon J. Bennett, Matthieu Lenoir, Frederik J. A. Deconinck.

**Formal analysis:** Silke De Waelle, Simon J. Bennett, Mark A. Scott, Frederik J. A. Deconinck.

**Investigation:** Silke De Waelle.

**Methodology:** Silke De Waelle, Frederik J. A. Deconinck.

**Project administration:** Silke De Waelle, Frederik J. A. Deconinck.

**Supervision:** Simon J. Bennett, Matthieu Lenoir, Frederik J. A. Deconinck.

**Writing – original draft:** Silke De Waelle, Simon J. Bennett, Mark A. Scott, Matthieu Lenoir, Frederik J. A. Deconinck.

**Writing – review & editing:** Silke De Waelle, Simon J. Bennett, Mark A. Scott, Matthieu Lenoir, Frederik J. A. Deconinck.

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
