## [Decision Letter · Decision Letter 0]

16 May 2025

Dear Dr. Bennett,

Thank you for submitting your manuscript to PLOS ONE. After careful consideration, we feel that it has merit but does not fully meet PLOS ONE’s publication criteria as it currently stands. Therefore, we invite you to submit a revised version of the manuscript that addresses the points raised during the review process.

**ACADEMIC EDITOR:**

Dear authors, thank you for sending your manuscript to Plos One for review. After peer review, we ask that you follow the suggestions made by the reviewers. Changes to the text must be identified, and a point-by-point response letter must be sent to each reviewer. I await your submission of the updated manuscript.

Best regards

We look forward to receiving your revised manuscript.

Kind regards,

Gustavo De Conti Teixeira Costa, Ph.D

Academic Editor

PLOS ONE

Journal Requirements:

Reviewers' comments:

Reviewer's Responses to Questions

**Comments to the Author**

1. Is the manuscript technically sound, and do the data support the conclusions?

Reviewer #1: Yes

Reviewer #2: Yes

2. Has the statistical analysis been performed appropriately and rigorously?

Reviewer #1: Yes

Reviewer #2: Yes

3. Have the authors made all data underlying the findings in their manuscript fully available?

Reviewer #1: Yes

Reviewer #2: Yes

4. Is the manuscript presented in an intelligible fashion and written in standard English?

Reviewer #1: Yes

Reviewer #2: Yes

Reviewer #1: Dear Editor-in-Chief,

I would like to express my gratitude for the opportunity to review this manuscript submitted to your journal. After a careful evaluation, I would like to suggest some improvements that could further strengthen the manuscript and enhance its clarity and scientific rigor, particularly in the introduction, methods, and discussion sections.

Introduction:

The introduction is well-directed and successfully presents the research problem, supported by relevant studies. However, to further strengthen this section, I recommend including references that specifically address the terminology "fast ball sports" and the necessity to process and monitor multiple sources of information from opponents, teammates, and the ball. Adding such references would substantiate the use of this nomenclature.

Additionally, in lines 45-47, where it is stated that on-field training for decision making is not simple, not individualized, and often overlooked in favor of technical skills training, it would be beneficial to include references that have discussed or demonstrated a similar focus on technical skills at the expense of decision-making training.

Methods:

In the methods section, particularly in lines 138-139, where it is mentioned that the number of opponents, court size, number of zones, and number of clips per condition are presented in Table 1, I suggest adding a brief explanation of how the court sizes were determined. This would clarify the rationale behind the experimental setup.

Furthermore, in lines 166-167, where it is mentioned that the aim was to ensure high-quality passes that would provide all spiking options, it would be important to briefly define what was considered a "high-quality pass" to enhance the understanding of the experimental conditions.

In lines 211-213, where different topics related to difficulty levels are discussed—such as the position of the block relative to the sidelines and the defenders relative to the block—it would be helpful to specify more clearly what criteria were used to assess these positions and what was considered a "bad pass."

Similarly, in lines 214-216, where the intervention aims to improve decision making in both standard and non-standard situations (e.g., bad pass or incorrectly formed block), a clearer definition of what constitutes a "correctly formed block" would strengthen the methodology description.

Finally, in lines 233-235, where a panel of three expert coaches selected the most optimal zone(s) for decision making, I recommend clarifying the criteria used to determine the "most optimal zone(s)."

Discussion:

Regarding the discussion section, I suggest restructuring it to first state that this was a pilot study, helping readers to appropriately contextualize the findings. Furthermore, I recommend organizing the discussion based on the hypotheses presented at the end of the introduction, using them as a framework to guide the interpretation of results.

Additionally, it would be beneficial to clearly separate the suggestions for future research and the limitations of the current study into distinct final sections, concluding with a concise summary of the key findings and contributions.

I believe these recommendations will significantly enhance the manuscript’s clarity, coherence, and scientific contribution. I appreciate your attention to these suggestions and remain at your disposal for any further clarifications.

Sincerely,

Reviewer #2: Review Comments

This study explored the impact of video training on the on-field decision-making ability of adolescent volleyball players, filling the gap in existing literature regarding the adolescent population and the effect of training transfer. The research design was reasonable, the hypotheses were clear, and the conclusion provided preliminary evidence for the application of video training in the improvement of adolescent sports skills, with practical value.

Problems and Suggestions

The initial sample size was small (N=20), and only 16 people completed the study in the end. This may affect the generalizability of the results.

The article mentions that the original plan was to have N=50, but due to difficulties in actual recruitment, only N=20 was achieved. It is necessary to explain whether a post hoc power analysis was conducted to assess whether the current sample size is sufficient to support the conclusion.

Was the training content of the control group completely unrelated to decision-making ability? It is necessary to provide additional explanations regarding the design logic of the tasks in the control group to avoid potential confounding factors that may indirectly affect decision-making ability.

The article mentions "random grouping by teams", but does not specify whether there are any baseline differences among the teams.

Although the on-site tests simulate the competition scenarios, they rely on a fixed defensive mode, which may differ from the dynamic complexity of real competitions.

Only the retention effect after 4 weeks was evaluated, and there was a lack of long-term follow-up.

The condition checks for the prerequisite of variance analysis were not mentioned.

Should multiple comparisons be corrected? The article mentioned using Bonferroni correction, but it is necessary to clarify which analyses it is specifically applied to.

**Do you want your identity to be public for this peer review?** For information about this choice, including consent withdrawal, please see our Privacy Policy

Reviewer #1: No

Reviewer #2: No

---

## [Author Response · Author response to Decision Letter 1]

16 Jun 2025

We have provided a point-by-point response to the reviewer's comment in an attached document

---

## [Decision Letter · Decision Letter 1]

4 Aug 2025

Dear Dr. Bennett,

Thank you for submitting your manuscript to PLOS ONE. After careful consideration, we feel that it has merit but does not fully meet PLOS ONE’s publication criteria as it currently stands. Therefore, we invite you to submit a revised version of the manuscript that addresses the points raised during the review process.

**ACADEMIC EDITOR:**

Dear author, I hope this email finds you well. Please proceed with the changes suggested by the reviewers. I await your receipt of the revised version.

Best regards

We look forward to receiving your revised manuscript.

Kind regards,

Gustavo De Conti Teixeira Costa, Ph.D

Academic Editor

PLOS ONE

Journal Requirements:

Reviewers' comments:

Reviewer's Responses to Questions

**Comments to the Author**

Reviewer #2: All comments have been addressed

Reviewer #3: (No Response)

2. Is the manuscript technically sound, and do the data support the conclusions?

Reviewer #2: Yes

Reviewer #3: Yes

3. Has the statistical analysis been performed appropriately and rigorously?

Reviewer #2: Yes

Reviewer #3: Yes

4. Have the authors made all data underlying the findings in their manuscript fully available?

Reviewer #2: Yes

Reviewer #3: Yes

5. Is the manuscript presented in an intelligible fashion and written in standard English?

Reviewer #2: Yes

Reviewer #3: Yes

Reviewer #2: 1. The planned N=50 (L94-96), but only N=16 (8 people per group) was actually completed. Although a post hoc sensitivity analysis was mentioned, the risk of insufficient power or effect size was not reported in the main text.

2. The review response stated that an error in the SPSS correction was found, which changed the p-value for verbal decision-making from 0.009 to 0.085, but the main text was not updated.

3. "On-field" should be consistent throughout the text (used as "on-field" in the abstract and L27, but as "field test" in L166).

Reviewer #3: 1. The keyword section should not repeat words used in the manuscript title.

2. Since the authors describe this study as a pilot study, it would be appropriate to reflect this in the title. Using a phrase such as "Pilot Study" would clarify the preliminary nature of the findings and properly define expectations regarding scope, sample size, and generalizability of the results.

3. The sample size (n = 20, of which only 16 completed the study) appears rather limited. Given the goal of assessing the effectiveness of video training on decision-making in young volleyball players, a larger sample size would strengthen both the validity and generalizability of the results.

4. The introduction lacks a justification for why decision-making is particularly important in volleyball, especially for female players, given the sample studied. Furthermore, the literature review lacks sufficient references to previous research on decision-making in volleyball.

5. The "Participants" section lacks detailed information about the participants, such as their age, gender, and training experience.

6. The manuscript does not provide information about the participants' stage of annual training cycle. This is particularly important for individuals recruited from different sports clubs, as changes in training load or competition phase could have influenced both cognitive and physical performance.

7. Conducting the home video test without researcher supervision may have introduced uncontrolled variables. A brief discussion of how this could have affected the reliability of the data would increase the study’s transparency. To what extent can the authors be certain that the test was performed by the intended participants rather than by an individual not involved in the study?

8. The manuscript does not specify whether a warm-up was conducted before the on-field assessment. If a warm-up was included, please provide detailed information regarding its structure and duration.

9. Regarding lines 136–140: It would be helpful to indicate whether the difficulty levels of the tasks were previously verified to ensure their appropriateness for the target age group.

10. Additional information regarding the environment in which the video-based decision-making task was conducted would be helpful. Was it conducted individually? In a quiet environment? Were any steps taken to limit distractions or prevent interactions between participants during the test? Clarifying these aspects would help assess the internal validity of the results. If the test was conducted in sessions, it would also be helpful to describe how other participants were treated during the individual assessments.

11. The manuscript suggests that the scenarios used in the on-court decision-making assessment mirrored those used in the video-based task. If so, there is a risk that the improvement may be due in part to memory or familiarity rather than to actual skill transfer. Although using adapted scenarios increases consistency, participants may have simply recalled the correct answers, especially if the post-test was conducted soon after the intervention.

12. The on-court test included a total of 40 attack trials from both the left and right sides of the net. The manuscript does not indicate whether rest periods were provided between trials. Clarifying this is important because a lack of rest periods can lead to fatigue, potentially impacting decision-making ability.

13. Although the manuscript states that cameras were positioned at various angles, the specific angles are not specified. Including this information would improve the clarity and repeatability of the on-field data collection procedure. Furthermore, there is no information about the frequency with which the cameras were recorded or the equipment used for this purpose.

14. The manuscript does not specify the height of the net used during the on-field assessment or the type of ball used. Providing this information is important for understanding whether the test conditions were appropriate for the participants' age group and level of competition.

15. The manuscript does not clearly indicate when the retention test was conducted in relation to the end of the 4-week training intervention. It would be helpful to specify whether it took place immediately after the last session or after a longer period. Clarifying the exact time (e.g., hours or days after the intervention) would improve the interpretation of retention effects.

16. It is unclear whether participants who were not actively performing the on-field task had the opportunity to observe others during the assessment. If so, this may have influenced their performance through observational learning or additional pressure. Clarifying whether the tests were administered individually or in isolation will help assess the consistency and reliability of the results.

**Do you want your identity to be public for this peer review?** For information about this choice, including consent withdrawal, please see our Privacy Policy

Reviewer #2: No

Reviewer #3: No

---

## [Author Response · Author response to Decision Letter 2]

8 Nov 2025

Reviewer #2:

1. The planned N=50 (L94-96), but only N=16 (8 people per group) was actually completed. Although a post hoc sensitivity analysis was mentioned, the risk of insufficient power or effect size was not reported in the main text.

We have added comment on this issue in the limitations section (lines 395-409), which now reads as below:

While this pilot study provides valuable insights into the use of individualized video-based training, it is important to acknowledge that there are some limitations. As noted in the methods, we initially planned to recruit 50 participants, but were only able to recruit 20 participants, of whom 4 did not complete the retention test, leaving us with 8 participants per group. Although this meant a reduction in power to detect smaller and moderate effects, a post-hoc sensitivity analysis indicated that this sample size was sufficient to detect an effect size of at least 𝜂p² = 0.175 with a power of 80%. Notably, the effect size of 𝜂p² = 0.04 found for motor execution was lower than both the a-priori and post-hoc critical thresholds, thus indicating that the finding of no significant improvement in this dependent measure by the experimental group was not due to insufficient power. As with some previous studies [20,28], the small sample size not only highlights the difficulty recruiting participants for training studies involving several sessions over several weeks (16 in total) followed by a delayed-retention, in which there are measures of off-field and on-field performance, but also raises a question about generalizability of the findings and the need for a replication study with a larger sample.

2. The review response stated that an error in the SPSS correction was found, which changed the p-value for verbal decision-making from 0.009 to 0.085, but the main text was not updated.

We are a bit confused as the main text was updated in the previous manuscript. This has been highlighted in the revised manuscript (line 331)

3. "On-field" should be consistent throughout the text (used as "on-field" in the abstract and L27, but as "field test" in L166).

This has been checked and amended where necessary.

Reviewer #3:

1. The keyword section should not repeat words used in the manuscript title.

We have amended the keywords such that they align with the most relevant subject areas taxonomies.

2. Since the authors describe this study as a pilot study, it would be appropriate to reflect this in the title. Using a phrase such as "Pilot Study" would clarify the preliminary nature of the findings and properly define expectations regarding scope, sample size, and generalizability of the results.

We appreciate the reviewers point and have modified the title as follows:

“Improving on-field decision making using video-based training – a pilot study with young volleyball players”

3. The sample size (n = 20, of which only 16 completed the study) appears rather limited. Given the goal of assessing the effectiveness of video training on decision-making in young volleyball players, a larger sample size would strengthen both the validity and generalizability of the results.

The reviewer makes a valid point, which we previously commented upon. On lines 108-111 of the revised manuscript we mention why we decided to proceed with the study:

Despite only being able to recruit 20 participants, it was decided to proceed with this sample on the basis that the initial effect size estimation could have been too conservative compared to previous similar work [12,13,18], and that any findings could still contribute to subsequent meta-analysis.

On lines 395-409, we have elaborated on the power of our model and warned for caution regarding the generalisability of the results. This section now reads:

While this pilot study provides valuable insights into the use of individualized video-based training, it is important to acknowledge that there are some limitations. As noted in the methods, we initially planned to recruit 50 participants, but were only able to recruit 20 participants, of whom 4 did not complete the retention test, leaving us with 8 participants per group. Although this meant a reduction in power to detect smaller and moderate effects, a post-hoc sensitivity analysis indicated that this sample size was sufficient to detect an effect size of at least 𝜂p² = 0.175 with a power of 80%. Notably, the effect size of 𝜂p² = 0.04 found for motor execution was lower than both the a-priori and post-hoc critical thresholds, thus indicating that the finding of no significant improvement in this dependent measure by the experimental group was not due to insufficient power. As with some previous studies [20,28], the small sample size not only highlights the difficulty recruiting participants for training studies involving several sessions over several weeks (16 in total) followed by a delayed-retention, in which there are measures of off-field and on-field performance, but also raises a question about generalizability of the findings and the need for a replication study with a larger sample.

4. The introduction lacks a justification for why decision-making is particularly important in volleyball, especially for female players, given the sample studied. Furthermore, the literature review lacks sufficient references to previous research on decision-making in volleyball.

We have included additional comment on the importance of decision-making in volleyball, as well as in youth athletes. We have also included some additional references to previous research on decision-making in volleyball. This additional detail has mainly been included towards the end of the introduction, where we start to focus on the task of volleyball. Before this, we refer more generally to various sports tasks that have been used when studying decision making through video-based training. In line with this more general focus, we have modified the title as follows: “Improving on-field decision making using video-based training – a pilot study with young volleyball players”.

Lines 66-68:

With youth athletes there are a similarly small number of studies that have investigated the effect of video-based training on decision making skills [17]. As with the adult studies, findings are mixed and little consideration is given to on-field transfer.

Lines 77-81:

More positive effects initially seem apparent in a recent meta-analysis on the effectiveness of decision making training in volleyball players [19]. However, closer inspection of the six original articles that studied youth players indicates that only four used off-field video-based training, and that of those only one found a 20% positive transfer to on-field performance [20].

Lines 86-94:

As commented above, decision making in volleyball requires the athlete to process and monitor multiple sources of information from opponents, teammates and the ball. The constant movement of the ball (i.e., it is not allowed to be caught and held) and the speed at which it travels (e.g., in excess of 100 km/h), combined with the dynamic player positioning, means that quick, accurate decisions on where and how to spike the ball are essential to capitalize on scoring opportunities. Simple and efficient training methods that improve decision making of youth athletes could have meaningful impact on their enjoyment and progression, as well as match outcome.

5. The "Participants" section lacks detailed information about the participants, such as their age, gender, and training experience.

We think the reviewer might have missed the information that was included in Table 2.

6. The manuscript does not provide information about the participants' stage of annual training cycle. This is particularly important for individuals recruited from different sports clubs, as changes in training load or competition phase could have influenced both cognitive and physical performance.

We have clarified in the revised manuscript (lines 221-222) that all players were recruited in the preseason of 2020-2021 and that the intervention started in week 1 of the season. The intention was that the intervention would supplement the conventional training sessions. However, due to the covid19 restrictions, physical training or games were prohibited, during the entire period of the study.

7. Conducting the home video test without researcher supervision may have introduced uncontrolled variables. A brief discussion of how this could have affected the reliability of the data would increase the study’s transparency. To what extent can the authors be certain that the test was performed by the intended participants rather than by an individual not involved in the study?

As stated in the original manuscript, the time and duration of video-based training was checked in the log-files created by the OpenSesame software, which the participant sent by email to the research team after each training session. While it is feasible that someone other than the participant completed the training, this is highly unlikely as they would need access to the participants computer and email account. We have included comment on this in the revised manuscript on lines 229-231.

8. The manuscript does not specify whether a warm-up was conducted before the on-field assessment. If a warm-up was included, please provide detailed information regarding its structure and duration.

Thank you for pointing this out. We have added this to the methods, on lines 196-198.

Prior to the test the participant was given time 15 minutes to warm-up, including light jogging, dynamic stretches and volleyball-specific passing and attacking drills.

9. Regarding lines 136–140: It would be helpful to indicate whether the difficulty levels of the tasks were previously verified to ensure their appropriateness for the target age group.

The test was based on earlier work by De Waelle et al. (2021) which provides detailed information on the development and validity of the test.

10. Additional information regarding the environment in which the video-based decision-making task was conducted would be helpful. Was it conducted individually? In a quiet environment? Were any steps taken to limit distractions or prevent interactions between participants during the test? Clarifying these aspects would help assess the internal validity of the results. If the test was conducted in sessions, it would also be helpful to describe how other participants were treated during the individual assessments.

The test took place in a separate and quiet room, with only the participant and experimenter present. This information has been added on lines 164-165.

11. The manuscript suggests that the scenarios used in the on-court decision-making assessment mirrored those used in the video-based task. If so, there is a risk that the improvement may be due in part to memory or familiarity rather than to actual skill transfer. Although using adapted scenarios increases consistency, participants may have simply recalled the correct answers, especially if the post-test was conducted soon after the intervention.

Our use of the word “replication” is potentially misleading. We have revised this section of the manuscript to clarify that the simulated offensive sequences used actual players and was intended to closely reproduce the video-based test. As for the issue of familiarity, we have revised the limitations section, which now reads (lines 411-416):

“It could also be interesting to include more sequences of offensive play to better reflect the dynamic complexity of real competitions, and to determine transfer to the field in actual volleyball matches. Indeed, our use of simulated 6x6 offensive sequences with actual players in the on-field test was intended to closely reproduce the video-based test, and as such did not allow us to determine transfer to novel attacking sequences.”

12. The on-court test included a total of 40 attack trials from both the left and right sides of the net. The manuscript does not indicate whether rest periods were provided between trials. Clarifying this is important because a lack of rest periods can lead to fatigue, potentially impacting decision-making ability.

We did not administer rest between trials, other than the preparation for the next trial, which reflects an actual game situation. The entire on-field test lasted 15-20 minutes, and we did not observe any obvious signs or reports of fatigue from the participants. This detail has been added at lines 212-215.

Rest periods between trials were not included, other than the time it took to prepare for the next trial, which reflects an actual game situation. The entire on-field test lasted 15-20 minutes, and we did not observe any obvious signs or reports of fatigue from the participants.

13. Although the manuscript states that cameras were positioned at various angles, the specific angles are not specified. Including this information would improve the clarity and repeatability of the on-field data collection procedure. Furthermore, there is no information about the frequency with which the cameras were recorded or the equipment used for this purpose.

We do not have the specific angles of the cameras relative to the court. We positioned the cameras such that the participant and zones on the opposite side of the court were clearly visible in the field of view. This was sufficient to enable experienced volleyball coaches to determine: 1) the direction in which the ball was played, in combination with the technique used, to decide on the player’s intended action; 2) what participants ended up doing based on the actual ball landing location, as indicated by the interrater reliability analysis (Cohen's kappa = 0.814). This detail has been added to the revised manuscript (lines 281-284)

14. The manuscript does not specify the height of the net used during the on-field assessment or the type of ball used. Providing this information is important for understanding whether the test conditions were appropriate for the participants' age group and level of competition.

Thank you for noting this. We have clarified on lines 192-193 of the revised manuscript that we used a Mikasa V020W size 5 ball and the net height was 2.24m, which is the regulation height for female players,

15. The manuscript does not clearly indicate when the retention test was conducted in relation to the end of the 4-week training intervention. It would be helpful to specify whether it took place immediately after the last session or after a longer period. Clarifying the exact time (e.g., hours or days after the intervention) would improve the interpretation of retention effects.

We have clarified that the the time in between the last training session and the retention test varied between 28 and 31 days (lines 132-134).

16. It is unclear whether participants who were not actively performing the on-field task had the opportunity to observe others during the assessment. If so, this may have influenced their performance through observational learning or additional pressure. Clarifying whether the tests were administered individually or in isolation will help assess the consistency and reliability of the results.

All tests were performed individually. This detail has been added to the revised manuscript on lines 215-216.

---

## [Decision Letter · Decision Letter 2]

24 Nov 2025

Improving on-field decision making using video-based training – a pilot study with young volleyball players

PONE-D-25-11115R2

Dear Dr. Bennett,

We’re pleased to inform you that your manuscript has been judged scientifically suitable for publication and will be formally accepted for publication once it meets all outstanding technical requirements.

Kind regards,

Gustavo De Conti Teixeira Costa, Ph.D

Academic Editor

PLOS ONE

Additional Editor Comments (optional):

Dear authors, thank you for submitting your manuscript to PLOS ONE. Congratulations on its acceptance, and I hope your manuscript will be cited by the international scientific community.

Sincerely,

Gustavo De Conti

Reviewers' comments:

Reviewer's Responses to Questions

**Comments to the Author**

Reviewer #2: All comments have been addressed

Reviewer #3: All comments have been addressed

2. Is the manuscript technically sound, and do the data support the conclusions?

Reviewer #2: Yes

Reviewer #3: Yes

3. Has the statistical analysis been performed appropriately and rigorously?

Reviewer #2: Yes

Reviewer #3: Yes

4. Have the authors made all data underlying the findings in their manuscript fully available?

Reviewer #2: Yes

Reviewer #3: Yes

5. Is the manuscript presented in an intelligible fashion and written in standard English?

Reviewer #2: Yes

Reviewer #3: Yes

Reviewer #2: This revised paper meets the requirements for publication. The author has supplemented the data and provided a more in-depth analysis of the results. The added content is in line with the modifications made to the proposed questions. Therefore, this paper can be published.

Reviewer #3: I particularly appreciate that the authors clearly identified the work as a pilot study in the title and throughout the text. This reflects the limited sample size, exploratory nature of the analyses, and limited generalizability of the results. In this format, the manuscript presents a coherent and transparent description of preliminary research.

The results should be interpreted with due caution and primarily as a basis for future studies on a larger scale.

The additional methodological clarifications (e.g., testing procedures, training period, ball and net specifications, warm-up, timing of the retention test) and the expanded discussion of limitations and future directions make the study suitably robust and informative for a pilot design.

**Do you want your identity to be public for this peer review?** For information about this choice, including consent withdrawal, please see our Privacy Policy

Reviewer #2: No

Reviewer #3: No

---

## [Editor Report · Acceptance letter]

PONE-D-25-11115R2

PLOS ONE

Dear Dr. Bennett,

I'm pleased to inform you that your manuscript has been deemed suitable for publication in PLOS ONE. Congratulations! Your manuscript is now being handed over to our production team.

Kind regards,

on behalf of

Dr. Gustavo De Conti Teixeira Costa

Academic Editor

PLOS ONE